# The Impact of a Nutritional Intervention Program on Eating Behaviors in Italian Athletes

**DOI:** 10.3390/ijerph18147313

**Published:** 2021-07-08

**Authors:** Annalisa Terenzio, Alice Cassera, Adriano Gervasoni, Alessandra Pozzi, Antonina Orlando, Andrea Greco, Paola Palestini, Emanuela Cazzaniga

**Affiliations:** 1School of Medicine and Surgery, University of Milano-Bicocca, Via Cadore, 48, 20900 Monza, Italy; info@annalisaterenzio.com (A.T.); alis.cassera@gmail.com (A.C.); pozzi.alessandra@outlook.it (A.P.); antonina.orlando@unimib.it (A.O.); paola.palestini@unimib.it (P.P.); 2Department of Human and Social Sciences, University of Bergamo, 24129 Bergamo, Italy; adrianogervasoni@hotmail.it (A.G.); andrea.greco@unibg.it (A.G.); 3Bicocca Center of Science and Technology for Food, University of Milano-Bicocca, Piazza della Scienza, 2, 20126 Milano, Italy

**Keywords:** nutritional intervention, athletes, lifestyle, nutrition, health

## Abstract

A balanced diet is a fundamental component of athletes’ health, training, and performance. The majority of athletes choose adequate quantities of macronutrients but, at the same time, do not respect World Health Organization dietary guidelines, eating a lot of discretionary food and not drinking enough water. Athletes need more nutritional education to improve the quality of their food choice. By modifying their eating habits, they could also enhance their performance. Our study aimed to evaluate the impact of nutritional intervention on eating habits in a group of Northern Italian athletes. A sample of 87 athletes (41 males and 46 females) aged 16.5 ± 2.9 was enrolled. We organized meetings and detected eating habits (before and after the meetings) using a food frequencies questionnaire. We found that nutritional intervention positively affected participants consumption of vegetables (*p* < 0.05), nuts (*p* < 0.001), legumes (*p* < 0.001), and fish (*p* < 0.05). Other aspects of the athletes’ eating habits were not significantly improved. Some gender differences were found; males increased their consumption of vegetables (*p* < 0.05) and nuts (*p* < 0.001), while females increased their intake of legumes (*p* < 0.001). Our finding suggested that nutritional intervention could promote healthy eating habits among athletes. If sports nutrition experts, coaches, personal trainers, sports medicine experts, and athletes cooperated, they could guarantee athletes’ health status.

## 1. Introduction

Eating habits are defined as how a person or group eats in terms of choices and quantity, and timing; World Health Organization (WHO) dietary guidelines highlight the importance of maintaining a normal Body Mass Index (BMI), eating at least five portions of fruit and vegetables per day, drinking lots of clean water, and limiting intake of sugars [1]. Reference intake Levels of Nutrients and Energy (LARN) reports the nutritional and health guidelines for the Italian population for people in good health.

A balanced diet is a fundamental aspect of athletes’ health, training, and performance. Moreover, nutrient timing has been shown to affect athletic performance. Nutrient timing incorporates the use of systematic planning that explains how to consume whole food, fortified food, and dietary supplements when it is necessary [2]. Athletes have the tendency not to meet general guidelines in terms of food variety and daily servings of food groups [3]. Nutrition periodization has been emerging as a key construct to optimizing sport-specific nutrition recommendations. The optimal amount and the type of food intake are considerably related to each athlete in conjunction with the specific training and sports demand [4].

Athletes’ nutritional analysis showed that the majority chose adequate quantities of macronutrients, but, at the same time, they included a lot of discretionary food. They did not have fruit or dairy products in their choice. In addition, apart from food intake, hydration should also be taken into consideration. It is reported that personalized and periodized hydration could optimize performance and guarantee the safety of athletes during the activity [5]. According to the guidelines about fluid replacement for performance optimization and dehydration prevention published by the American College of Sport Medicine, there is considerable variability among individuals, environmental conditions, and different physical activities [6,7]. Many athletes report that they consume foods with little nutritional value and do not eat and drink sufficiently for practice and competition [8].

The athlete needs to be motivated to follow an individualized nutritional plan to optimize either race-day nutrition or significant championship nutrition choice [4]. Additionally, when examining gender differences in dietary and nutrition knowledge, results are equivocal between the genders [9], with some reporting better outcomes for women [10]. Published research regarding women volleyball athletes has shown that educational sessions can improve nutritional knowledge and dietary intake [8]. Given these assumptions, athletes need more nutritional education to improve their choices [3].

The purpose of this study is to evaluate the effect of nutritional intervention on a group of Northern Italian athletes’ eating behavior in terms of eating habits and physical status, in particular: (a) to observe possible changes in food consumption; (b) to assess possible changes in the athletes’ health status in terms of BMI, adiposity, etc.; (c) to assess differences between male and female athletes in the process.

When assessing body composition in the sports field, different parameters (fat mass, intra/extra cellular water or muscle mass) should be evaluated [11]. In fact, several studies have shown a relationship between BMI and performance [12,13,14]. Still, the BMI value by itself is not sufficient to define the health status of athletes with greater muscle mass. For this reason, the use of the WtHR should be considered for the estimation of abdominal adiposity [15,16].

It was hypothesized that nutritional intervention would be beneficial with benefits to the assessed outcomes.

## 2. Materials and Methods

### 2.1. Study Design

A composite team comprised scientists and nutritionists was formed. The study was conducted between April 2018 and July 2019.

After the signing of the informed consent, all participants (parents for minors) were invited to fill out a questionnaire (baseline, T1). Next, correct eating habits were explained during an athlete training meeting. The meeting was held by health professionals (nutritionists) of the University of Milano-Bicocca. After 3 months (T2) from the training meeting, a second training meeting was organized for the same subjects. The purpose of the second meeting was to provide specific nutrition guidelines for athletes. After 3 months (follow-up, T3) from the second training meeting, the same questionnaire was reconfirmed to evaluate the assimilation of information and the general improvement of athletes’ habits.

During the first meeting, parents and athletes were counselled on the importance of having five meals per day, in particular having a morning and afternoon snack, the role of daily consumption of fruit and vegetables, the suggested amount of daily water intake, and the importance of reducing the consumption of sweetened beverages and sugary snacks.

During the second meeting the health experts explained how to organize daily food choices, the best time to snack considering the training schedule, what foods to choose for snacks and main meals, and when the use of supplements is necessary.

Data were collected with a questionnaire (Appendix A) that was filled either by the athletes or by their parents or guardians (if the athletes were minors) at the baseline (T1) and follow-up (T3). The questionnaire was divided into two parts. The first part contained questions related to the interviewee’s daily habits, in particular the athletic discipline practiced, training times, where meals are eaten, and usual eating times in case of training/competitions.

The second part was related to eating habits, how many meals the subject consumes during the day, and food consumption frequencies. For example, for fruit, we asked how often on average they consumed it, with six possible responses: 3/4 portions/day, 2 portions/day, 1 portion/day, 1 portion/3–4 time per week, 1 portion/3–4 time per month, never.

### 2.2. Participants

Before inclusion in the study, participants were informed about the research and its purpose, were invited to ask any questions, and then provided their informed consent. The study was conducted according to the Declaration of Helsinki guidelines, and approved by the Institutional Review Board of the University of Milano-Bicocca (479, 08/10/19).

One hundred and twenty five Italian athletes (ages 13–25 years) were recruited, but only 87 athletes participated in the study. The recruited athletes practiced disciplines such as jumping, 100–200 m distance running, and 300 m or more distance running.

### 2.3. Procedures

Study participants attended an anthropometric measurement. Height (Seca 206, Hamburg, Germany) and weight (Seca 703, Hamburg, Germany) were measured in underwear. Weight was approximated to hectograms, while height precision was 5 mm. The WHO’s BMI classification (2020) was divided into underweight, normal, overweight, and obese.

Waist circumference was measured to the nearest 0.1 cm by a nonelastic flexible tape in the standing position following expiration in triplicate at a level midway between the lower costal border and the top of the iliac crest, and the average used for subsequent analysis. From these measurements the waist-to-height ratio (WHtR) was calculated.

### 2.4. Statistical Analysis

The statistical analysis of the data was carried out using IBM^®^ SPSS^®^ Statistics 25 software (IMB, Armonk, NY, USA).

Preliminarily, descriptive statistics, in terms of frequencies and percentages for categorical variables and mean and standard deviations for continuous ones, were analyzed to assess the distribution of the collected variables. Subsequently, statistical analyses were implemented to verify the proposed objectives. In order to verify the possibility of statistical difference among measured variables between T1 and T3, paired sample *t*-test (for continuous data) and McNemar and Marginal Homogeneity tests (for categorical data) were used. The McNemar’s test refers to 2 × 2 contingency tables of a dichotomous character of balanced pairs of subjects to decide whether the frequencies of the row and column are equal or if there is a difference. The marginal homogeneity test is the test corresponding to McNemar for non-dichotomous qualitative variables; the extension of McNemar’s test arises in cases where independence does not automatically hold between the pairs (e.g., there are clusters of paired data where pairs in a cluster might not be independent, but there is independence between different clusters) [17]. An asymptotic significance *p*-value lower than 0.05 means that there was a significance in the change in intake of some foods. The analyses were carried out on the entire sample comparing T1 and T3 scores; in addition, these tests were also used separately on male and female subject groups in order to explore gender difference.

In relation to sample size estimation, the minimum expected count was based on widely used guidelines by Cochran [18] that suggest that, in a contingency table, the expected value for each cell has to be at least equal to 5; the total sample size corresponds to the total frequencies of each row and column. Since in this study there were variables with 6 categories, considering the main aim to compare the paired sample at T1 and T3, a sample size of 60 participants was needed.

## 3. Results

The questionnaire provided a high amount of information, but only the most significant is considered in this paper.

### Descriptive Statistics

Table 1 shows the descriptive statistics of the total sample at T1 and T3.

A total of 87 athletes participated in the study. The mean age was 16.5 years (SD = 2.9); regarding BMI, most participants presented an average weight. After the intervention, the group had significant slight increases in age, weight, height, BMI, WHtR; regarding BMI, there was a decrease of participants with normal weight and an increase in underweight participants.

Table 2 shows the descriptive statistics of the sample divided into males and females at T1 and T3. At T1, there were 41 males with an average age of 16.8 years (SD = 3.4); 21.95% of males at T1 were underweight, 68.4% normal weight, 9.7% overweight, and none were obese. After the intervention, males had significant slight increases in age, weight, height, BMI, WHtR; regarding BMI, a person who was underweight at T1 and a person who was overweight at T1 fell into the category of normal weight. There were 46 females present at T1 with an average age of 16.1 years (SD = 2.3); 19.5% of females at T1 were underweight, 78.2% normal weight, 4.6% overweight, and none were obese. After the intervention, females had significant slight increases in age, weight, height, BMI, and WHtR; no significant changes were observed in the category of BMI.

Regarding eating habits, Table 3 shows frequencies of consumption of fruit, vegetables, legumes, fish, meat, and nuts at T1 and T3 for all athletes and males and females separately.

Regarding vegetable consumption, there was a significant increase in athletes that never ate vegetables (*p* = 0.010). Throughout the study, there was an increase in daily vegetable consumption; at T3, all subjects ate vegetables at least once or twice a month. Concerning gender differences, this result was significant for males (*p* = 0.042), while it was not significant for females (*p* = 0.109). Throughout the study, there was an increase in the weekly consumption of vegetables.

Regarding legumes, there was a significant increase during the week considering the entire group (*p* < 0.001), males (*p* < 0.001), and females (*p* < 0.001) separately. The most significant increase in the consumption of legumes occurred in the female sample.

There was also a significant increase in fish consumption during the week, considering the whole group (*p* < 0.001), males (*p* < 0.001), and females (*p* = 0.012) separately. The results showed a general increase in fish consumption throughout the week and an increase in subjects that did not eat it before. Moreover, both males and females followed the same pattern: a general decrease in the response “1–2 per month”, and a general rise in the answers “3–4 times per week”, “1–2 times per week”, and “never”.

Regarding the consumption of nuts, there was a significant increase in the total sample (*p* = 0.001). Throughout the study, there was a general increase in weekly and monthly nuts consumption; at T3, all subjects ate vegetables at least once or twice a month. Furthermore, regarding gender differences, this result was significant for males (*p* = 0.001), while it was not significant for females (*p* = 0.118).

No significant differences between T1 and T3 were obtained for the consumption of fruit (*p* total sample = 0.18, *p* male sample = 0.43, *p* female sample = 0.28), meat (*p* total sample = 0.50, *p* male sample = 0.33, *p* female sample = 1.00), and water (*p* total sample = 1.00, *p* male sample = 1.00, *p* female sample = 1.00). The variable concerning water intake was measured by eight items; then, it was decided by convention to unify them to have a single item. The tests’ results regarding the water intake’s variables indicated a general intake more distributed during the day from T1 to T3; however, it was not significant.

## 4. Discussion

The purpose of our project was to evaluate the effectiveness of sports nutrition education interventions on subjects practicing athletics.

In particular, we investigated whether raising awareness about the adoption of a healthy and balanced diet had positive effects in terms of improving eating habits and body composition.

There are few published studies on nutritional interventions in athletes, and due to the different methodologies used, the results are inconsistent. Through the administration of the food questionnaire, we examined the eating habits of the athletes participating in the study. In particular, we investigated whether the frailties found in the general population were also present in the nutrition of athletes. These weaknesses are represented by low fruits and vegetables consumption, water, the bad habit of skipping breakfast, and excessive salt consumption [19,20].

Observing the results, we can see how the athletes, despite having a normal weight status, did not change between T1 and T3, showing eating habits characterized by the same frailties of the general population as previously reported [21,22].

Moreover, it is known that the nutritional knowledge of athletes and their coaches has been shown to be inadequate and it is strongly associated with healthy eating and healthy food choices [21]. For these reasons, carrying out food education interventions to improve daily habits can be of fundamental importance and impact [22].

Regarding fruits and vegetables consumption, at T1, the athletes did not follow recommendations. After the training, the athletes showed an improvement in vegetable consumption, particularly in the male group, as previously reported [22]. Athletes showed a higher intake of nuts at T3, particularly in the male group, probably used as a snack as recommended in meetings.

We observed an improvement in fish and legumes instead of meat consumption. Probably athletes understood the importance of different food choices.

We did not find differences between T1 and T3 regarding water intake. Despite the importance of athlete hydration, only a few studies [22,23] have investigated the effect of nutritional interventions on athlete hydration practices. Education alone did not change hydration behaviors [23]. The improvement in hydration status was only achieved in young athletes by individual prescriptions or increasing this nutrient [22]. Prescribed drinking volumes should be based on individual sweat rates, environmental conditions, and practices recommended by the National Athletic Trainers’ Association (NATA) [23].

We acknowledge some limitations: first of all, the use of athletes from only one sporting institution. Such limitations may include a survey’s susceptibility to recall bias and under- or over-reporting by subjects and we did not focus on the coaching staff. Although the coaches were present during the meeting, the information was directed toward the athletes rather than them. Another limitation is that we conducted two meetings that might have had a limited effect on changing eating behaviors in athletes and did not attempt to determine the baseline nutrition knowledge of the athletes or coaches. Furthermore, the use of more accurate methods to estimate body composition are missing, for example, bioelectric impedance analysis (BIA) and bioelectric impedance vector analysis (BIVA) that have recently gained attention in sports [11].

## 5. Conclusions

Thanks to our intervention, many athletes have improved their eating habits, positively impacting health but presumably also sports performance.

We believe that in order to be effective over time, a nutritional education intervention should consist of several meetings, be open to athletes, families, and coaches, and provide analysis of the knowledge that is the basis of food change.

Finally, sport nutrition experts, personal trainers, and sport medicine experts should collaborate to guarantee athletes’ health status.

## Figures and Tables

**Table 1 ijerph-18-07313-t001:** Anthropometric characteristics of athletes at T1 and T3.

	T1 (*n* = 87)		T3 (*n* = 87)		
	Mean	S.D.	Mean	S.D.	*p*-Value *
Age (years)	16.5	2.9	17.0	3.0	<0.001
Weight (kg)	58.2	11.4	61.2	11.8	<0.001
Height (cm)	168.9	9.1	170.3	9.1	<0.001
BMI (kg/m^2^)	20.2	2.5	21.0	2.6	<0.001
WHtR	0.4	0.03	0.4	0.03	<0.001
Underweight (%)	0.0		14.9		0.003
Normal (%)	94.3		79.3		
Overweight (%)	5.7		4.6		
Obese (%)	0		1.1		

* *p*-Value comparing results at baseline and post-intervention by paired sample *t*-test, McNemar and Marginal Homogeneity tests. BMI = Body mass index; WHtR = waist-to-height ratio.

**Table 2 ijerph-18-07313-t002:** Anthropometric characteristics of males and females at T1 and T3.

	T1				T3					
	Males (*n* = 41)		Females (*n* = 46)		Males (*n* = 41)		Females (*n* = 46)		*p*-Value *	
	Mean	S.D.	Mean	S.D.	Mean	S.D.	Mean	S.D.	Males	Females
Age (years)	16.8	3.4	16.1	2.3	17.4	3.5	16.6	2.5	<0.001	<0.001
Weight (kg)	61.7	13.4	55.0	8.1	65.7	13.6	57.2	8.1	<0.001	<0.001
Height (cm)	174.0	9.6	164.4	5.8	176.0	8.3	164.8	5.8	<0.001	0.004
BMI (kg/m^2^)	20.1	2.8	20.3	2.2	20.9	2.9	21.0	2.3	<0.001	<0.001
WHtR	0.4	0.03	0.4	0.03	0.4	0.03	0.4	0.03	0.009	0.008
Underweight (%)	21.9		19.5		19.5		10.8		0.003	0.257
Normal (%)	68.4		78.2		73.2		84.7			
Overweight (%)	9.7		4.6		7.3		2.17			
Obese (%)	0		0		0		2.17			

* *p*-Value comparing results at baseline and post-intervention by paired sample *t*-test, McNemar and Marginal Homogeneity tests. BMI = Body mass index; WHtR = waist-to-height ratio.

**Table 3 ijerph-18-07313-t003:** Fruit, Vegetables, Legumes, Fish, Meat, and Nuts Consumption at T1 and T3.

	All (*n* = 87)			Males (*n* = 41)			Females (*n* = 46)		
	T1 (%)	T3 (%)	*p*-Value *	T1 (%)	T3 (%)	*p*-Value *	T1 (%)	T3 (%)	*p*-Value *
Fruit									
3/4 portions/day	23.3	18.4	0.188	19.5	9.8	0.437	26.7	26.1	0.281
2 portions/day	27.9	27.6		26.8	24.4		28.9	30.4	
1 portion/day	18.6	27.6		17.1	29.3		20.0	26.1	
1 portion/3–4 times per week	19.8	16.1		22.0	17.1		17.8	15.2	
1 portion/3–4 times per month	8.1	8.0		12.2	17.1		4.4	0	
Never	2.3	2.3		2.4	2.4		2.2	2.2	
Vegetables									
2 times per day	33.3	34.1	0.010	19.5	24.4	0.042	45.7	43.5	0.109
1 time per day	23.0	27.3		24.4	24.4		21.7	28.3	
3–4 per week	16.1	15.9		24.4	22.0		8.7	10.9	
1–2 per week	11.5	18.2		9.8	22.0		13.0	15.2	
1–2 per month	6.9	4.5		9.8	7.3		4.3	2.2	
Never	9.2	0.0		12.2	0.0		6.5	0.0	
Legumes									
Everyday	1.1	6.8	<0.001	0.0	7.3	<0.001	2.2	4.3	<0.001
3–4 per week	12.6	14.8		14.6	12.2		10.9	17.4	
1–2 per week	40.2	59.1		36.6	53.7		43.5	65.2	
1–2 per month	24.1	0.0		24.4	0.0		23.9	0.0	
Never	21.8	19.3		24.4	26.8		19.6	13.0	
Fish									
Everyday	1.2	0.0	<0.001	0.0	0.0	<0.001	2.2	0.0	0.012
3–4 per week	10.5	20.7		9.8	17.1		10.9	23.9	
1–2 per week	55.8	60.9		48.8	56.1		60.9	63.0	
1–2 per month	24.4	0.0		29.3	0.0		19.6	0.0	
Never	8.1	18.4		12.2	24.4		4.3	13.0	
Meat									
2 times per day	3.6	4.7	0.500	5.1	9.8	0.336	2.2	0.0	1.000
1 time per day	40.5	29.4		51.3	31.7		31.1	27.3	
3–4 per week	36.9	47.1		33.3	41.5		40.0	52.3	
1–2 per week	17.9	17.6		10.0	17.1		24.4	18.2	
1–2 per month	0.0	1.2		0.0	0.0		0.0	2.3	
Never	1.2	0.0		0.0	0.0		2.2	0.0	
Nuts									
2 times per day	3.4	1.2	0.001	4.9	2.4	<0.001	2.2	0.0	0.118
1 time per day	11.5	9.3		4.9	12.2		17.4	6.5	
3–4 per week	8.0	18.6		9.8	12.2		6.5	23.9	
1–2 per week	18.4	44.2		12.2	36.6		23.9	47.8	
1–2 per month	37.9	26.7		36.6	34.1		39.1	19.6	
Never	20.7	0.0		31.7	0.0		10.9	0.0	

* *p*-Value comparing results at baseline and post-intervention by McNemar and Marginal Homogeneity tests.

## Data Availability

The datasets used and/or analyzed during the current study are available from the corresponding author on reasonable request.

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
