# Peer review of "The Impact of a Nutritional Intervention Program on Eating Behaviors in Italian Athletes"

_ijerph, 2021, doi:10.3390/ijerph18147313_

Round 1
Reviewer 1 Report
With this study, the authors aimed to assess the effect of nutritional education in young athletes. For that purpose, a questionnaire was used to assess the nutritional habits and two educational interventions performed. The paper addresses an interesting issue. Some concerns are listed below.
Introduction
When writing the introduction avoid using expressions like “in conclusion” because it can compromise the need to perform the study.
The introduction could include enlightening about the power of BMI to define health status in athletes and the benefit of using Waist to height ratio.
Methods
Please refer to how participants were recruited and how many participated in the study. Describe better the participants (type of athletes, modalities performed, level of competition, etc.).
Explain also if all the participants were evaluated on both moments (T1 and T3) and if there were any dropouts.
Please refer if the study was approved by any Ethics Committee.
The values presented are waist-to-height ratio (WHtR), please refer to it in the methods.
Revise the statistical description
Results
N should be added to the tables as the units for all variables, eg. BMI (kg/m2).
Table legend should include all abbreviations.
It would be nice to have a summary of the table 3 information in a graphic.
Include the statistical comparison results in table 3.
Discussion
Study limitations should be presented and their impact on the study discussed.
Avoid identifying the club where the study was performed, which compromises confidentiality.
Conclusions should be presented in the conclusion section.
Author Response
Reply to Reviewer #1
Introduction
When writing the introduction avoid using expressions like “in conclusion” because it can compromise the need to perform the study.
As suggested by reviewer 3, we have revised the introduction section. We replaced “in conclusion” with “give these assumption” (line 110)
The introduction could include enlightening about the power of BMI to define health status in athletes and the benefit of using Waist to height ratio.
We have inserted a sentence to explain BMI and WtHr (line 117-120) and 3 new references [16-18].
Methods
-Please refer to how participants were recruited and how many participated in the study. Describe better the participants (type of athletes, modalities performed, level of competition, etc.).
We have inserted a sentence to describe the participants and the sports disciplines (line 131-133)
-Explain also if all the participants were evaluated on both moments (T1 and T3) and if there were any dropouts.
We thank the reviewer for raising this point that give to us the opportunity to highlight better this point. As said before, all the statistical analyses were evaluated only with those participants who completed the program. No dropouts were present in the study.
-Please refer if the study was approved by any Ethics Committee.
We have inserted the ethical approval (line 129)
-The values presented are waist-to-height ratio (WHtR), please refer to it in the methods.
As suggest, we inserted a phrase in the methods (line 217)
-Revise the statistical description
As requested, we revised and tried to clarify the statistical analysis.
Results
-N should be added to the tables as the units for all variables, eg. BMI (kg/m2).
All the measurements were added to the tables.
-Table legend should include all abbreviations.
All the abbreviations were included into the tables’ legend.
-It would be nice to have a summary of the table 3 information in a graphic.
We thank again the reviewer for this point. Since it is also asked to add statistical comparison in the table 3, we decided to add graphics as supplementary material to have a different method to summarize the results (Figure 1/2/3/4/5/6).
-Include the statistical comparison results in table 3.
We added statistical comparison in the table 3.
Discussion
-Study limitations should be presented and their impact on the study discussed.
We thank again the reviewer for this point. We added the study limitation at the end of discussion (line 563-570).
-Avoid identifying the club where the study was performed, which compromises confidentiality.
As suggested we have delete the name of the club, we have left it the acknowledgement section
-Conclusions should be presented in the conclusion section.
As suggested we have revised the conclusion section.
As suggested by the Reviewer 1, we have revised the English language.
Reviewer 2 Report
The authors have made considerable modifications and answered all questions previously proposed.
In my opinion the paper can be accepted now.
Author Response
Reply to Reviewer #2
The authors have made considerable modifications and answered all questions previously proposed. In my opinion the paper can be accepted now.
Thank you for the acceptance.
Reviewer 3 Report
Abstract: descriptive statistics are missing.
Introduction:
- Introduction is organized in so many paragraphs. I already got lost in the 3rd paragraph. Please revise.
- Additionally, the rationale for the conduct of this study is not very clear. I, therefore, suggest revising the introduction to make clear why your study is of importance for the field.
Methods overall:
- Confused as written. It should be organized in: Study design; Participants, Procedures, and Statistical Analysis.
- Sample size is missing. I am blinded about the number and the descriptive characteristics of the participants, as well as inclusion and exclusion criteria.
- Please provide results for a power analysis that indicate your sample size was appropriate.
- For any equipment used to collect data that contributed to an analyzed variable, provide the model, manufacturer, and country, unless it is contained in the previously published methodology you have cited.
Discussion section
- The discussion section is very descriptive and offers limited comparisons to previous research. How do practitioners benefit from that? Authors are therefore encouraged to make substantial changes throughout to improve the overall quality. In the current form the rationale for the study is not clear, the new value is unclear, and I have difficulties finding specific take-home messages for practitioners.
Author Response
Reply to Reviewer #3
-Abstract: descriptive statistics are missing.
As suggested, we added a descriptive statistics in the abstract.
-Introduction:
- Introduction is organized in so many paragraphs. I already got lost in the 3rd paragraph. Please revise.
-Additionally, the rationale for the conduct of this study is not very clear. I, therefore, suggest revising the introduction to make clear why your study is of importance for the field.
We thank the reviewer for these observations. We have revised the introduction section
-Methods overall:
- Confused as written. It should be organized in: Study design; Participants, Procedures, and Statistical Analysis.
As suggested, we organized methods in 4 paragraphs
-Sample size is missing. I am blinded about the number and the descriptive characteristics of the participants, as well as inclusion and exclusion criteria.
As also suggested by reviewer 1, we have inserted a sentence to describe the participants and the sports disciplines (line 133-135). All the athletes of this society are been invited to participate, no exclusion criteria are been inserted.
- Please provide results for a power analysis that indicate your sample size was appropriate.
We added information about the appropriateness of the sample size considering guidelines suggested by Cochran (1954) (reference [20] in the revised manuscript) (line271).
-For any equipment used to collect data that contributed to an analysed variable, provide the model, manufacturer, and country, unless it is contained in the previously published methodology you have cited.
As suggested we added the data required (line 137).
-Discussion section
The discussion section is very descriptive and offers limited comparisons to previous research. How do practitioners benefit from that? Authors are therefore encouraged to make substantial changes throughout to improve the overall quality. In the current form the rationale for the study is not clear, the new value is unclear, and I have difficulties finding specific take-home messages for practitioners.
We thank the reviewer for these observation. We have revised the discussion and conclusion sections.
As suggested by the Reviewer 3, we have revised the English language
Round 2
Reviewer 1 Report
In my opinion, the rationale for using the Waist to Height ratio is not clear in the sentences added. Please add the references used to state “For this reason, the use of the WtHR should be considered for the 76 estimation of fat mass”.
125 Northern Italy athletes (ages 13-25 years) were recruited, but only 87 athletes 87 participated in the study. I don’t understand why the region of the athletes is referred. If there is a reason please describe it, otherwise, only the athletes’ description is needed.
Author Response
Reply to Reviewer #1
In my opinion, the rationale for using the Waist to Height ratio is not clear in the sentences added. Please add the references used to state “For this reason, the use of the WtHR should be considered for the 76 estimation of fat mass”.
We have modifies the sentence (lines 81-86) and added two new references [15, 16].
125 Northern Italy athletes (ages 13-25 years) were recruited, but only 87 athletes 87 participated in the study. I don’t understand why the region of the athletes is referred. If there is a reason please describe it, otherwise, only the athletes’ description is needed.
We have removed “northern”.
Reviewer 2 Report
The paper untitled “Effects of nutritional education on eating behaviors in a group of Italian Athletes” present a study about the effect of organized educational meetings and the detected eating habits of the participants by using questionnaire (before and after the meetings).
The paper presents interesting appointments, however some points must be revised.
My first point is related to the title. In my opinion the use of “nutritional education” must be reviewed, once this terms given us the impression about a real and robust education in nutrition, like a undergraduate course, or specialization, between others.
Line 19, at abstract should be revised. Which others aspects are authors talking about?
Line 29: WHO must be defined, as well as, BMI at line 30 and LARN at line 31.
The study was previously submitted to an ethical committee?
What is WHtR in table 1?
Results of Table 1 and table 2 must be placed at material and methods section, once they consist the material of the analyses of the study. Results obtained after the intervention, must be presented at another table or figure, once these are results.
Besides, the number of participants, what the authors think about this number?
Line 164: “Regarding vegetable consumption, there was a significant increase in the total sample (p=.010).” I cannot agree with this affirmation. Maybe the authros wanted to say that a significant increase was observed in relation to athletes which never ate vegetables. I think you must rewrite the sentence.
Line 175-176: I didn’t understand “an increase in non-food intake”? You need to specific it.
Line 176-78: you are talking about fishes yet? Be more specific.
Line 181: vegetable or nuts?
Line 194: why the authors do not present this information before? “on subjects practicing athletics in the club Atletica Bergamo 1959”
Line 198: “Through the administration of the food questionnaire, we examined the lifestyle”, no you didn’t exanimated lifestyle.
Line 207: “Moreover, the nutritional knowledge of athletes and their coaches was shown to be inadequate”. No, you cannot say this, the questionnaire didn’t evaluated the nutritional knowledge, the questionnaire evaluated eating habits.
Line 224: define NATA
I think the evaluation a long time must be also evaluated. In example, after one year of the intervention, have the athletes maintained the habits?
Author Response
Reply to Reviewer #2
The paper untitled “Effects of nutritional education on eating behaviors in a group of Italian Athletes” present a study about the effect of organized educational meetings and the detected eating habits of the participants by using questionnaire (before and after the meetings).
The paper presents interesting appointments, however some points must be revised.
-My first point is related to the title. In my opinion the use of “nutritional education” must be reviewed, once this terms given us the impression about a real and robust education in nutrition, like a undergraduate course, or specialization, between others.
As suggested, we have substituted “education” with “intervention” in the title and overall the manuscript.
-Line 19, at abstract should be revised. Which others aspects are authors talking about?
As suggested we have revised the abstract but data on meat, fruit, nuts and water are only in the manuscript for brevity.
-Line 29: WHO must be defined, as well as, BMI at line 30 and LARN at line 31.
As suggested, we have defined the acronyms.
-The study was previously submitted to an ethical committee?
We inserted the ethical approval (line 131).
-What is WHtR in table 1?
We added this info in the tables’ legend. Moreover, in the text of the manuscript there are now the explanation of the WHtR measure.
-Results of Table 1 and table 2 must be placed at material and methods section, once they consist the material of the analyses of the study. Results obtained after the intervention, must be presented at another table or figure, once these are results.
We respected indication used usually from many journals that consider the number of participants and the related descriptive information a results. Moreover, in relation to the comparison of data at T1 and T3, in our opinion, it is better to present the results obtained before and after the intervention in the same table; in this way, we can insert information related to the statistical comparisons (as for example requested in the table 3 from other reviewers). We hope this this choice can be understood.
-Besides, the number of participants, what the authors think about this number?
We added information of the number of participants invited to participate to the study (n=125); we recruited a sample of 87 participants and it means that 70% of people decided to participate. We think this is a good result for a study in which the participation is totally voluntary.
-Line 164: “Regarding vegetable consumption, there was a significant increase in the total sample (p=.010).” I cannot agree with this affirmation. Maybe the authors wanted to say that a significant increase was observed in relation to athletes which never ate vegetables. I think you must rewrite the sentence.
As suggested, we have changed the phrase (line 450).
-Line 175-176: I didn’t understand “an increase in non-food intake”? You need to specific it.
As suggested, we rephrased the sentence (line 464).
-Line 176-78: you are talking about fishes yet? Be more specific.
As suggested, we changed the phrase (line 464).
-Line 181: vegetable or nuts?
As suggested, we changed the phrase (line 469).
-Line 194: why the authors do not present this information before? “on subjects practicing athletics in the club Atletica Bergamo 1959”
As also suggested by reviewer 1, we have delete the name of the club, we have left it the acknowledgement section.
-Line 198: “Through the administration of the food questionnaire, we examined the lifestyle”, no you didn’t exanimated lifestyle.
As suggested, we have deleted “life style”
-Line 207: “Moreover, the nutritional knowledge of athletes and their coaches was shown to be inadequate”. No, you cannot say this, the questionnaire didn’t evaluated the nutritional knowledge, the questionnaire evaluated eating habits.
The phase is referred to Alaunyte et al 2015 [ref17] and it is not referred to our questionnaire; we rephrased the sentence.
-Line 224: define NATA
As suggested, we have defined the acronym.
-I think the evaluation a long time must be also evaluated. In example, after one year of the intervention, have the athletes maintained the habits?
We thank again the reviewer for this point. We added the study limitation at the end of discussion (line 567-574).
As suggested by the Reviewer 2, we have revised the English language
Reviewer 3 Report
Terenzio and co-workers improved their manuscript by addressing all the previous comments. However, I believe more effort is needed in order to achieve a high-quality standard as required by IJERPH.
Alternative title: The impact of a nutritional intervention program on eating behaviors in Italian athletes
Abstract: statistical data are missing.
Introduction and discussion: body composition is mentioned among the keywords, but the manuscript presents poor connections between nutritional status and body composition. A few sentences should be added in order to highlight which body composition parameters (e.g., fat mass, muscle mass, hydration) are more informative and related to health status and sports performance. Also, in the discussion section, a brief list of assessment methods (BIA, anthropometry, DXA) should be reported. See: Assessment of Body Composition in Athletes: A Narrative Review of Available Methods with Special Reference to Quantitative and Qualitative Bioimpedance Analysis. Nutrients. 2021 May 12;13(5):1620. doi: 10.3390/nu13051620. PMID: 34065984; PMCID: PMC8150618.
The second paragraph is not really necessary and not linked with the 1st and 3rd. If you do not want to remove it, you can switch it with the 3rd.
“Then, many athletes report to consume foods with little nutritional value and not to 59 eat and drink sufficiently for practice and competition [13].” A sentence cannot be reported as a paragraph; please revise.
“Given these assumptions, athletes need more nutritional education to improve their 67 choices [8].” A sentence cannot be reported as a paragraph; please revise.
In general, there are too many paragraphs making the reading boring and not very concise.
Methods:
Study design should be the first paragraph.
Author Response
Reply to Reviewer #3
Terenzio and co-workers improved their manuscript by addressing all the previous comments. However, I believe more effort is needed in order to achieve a high-quality standard as required by IJERPH.
Alternative title: The impact of a nutritional intervention program on eating behaviors in Italian athletes
We thank the reviewer for the suggestion, we have modified the title.
Abstract: statistical data are missing.
Statistical data have been inserted in the abstract.
Introduction and discussion: body composition is mentioned among the keywords, but the manuscript presents poor connections between nutritional status and body composition. A few sentences should be added in order to highlight which body composition parameters (e.g., fat mass, muscle mass, hydration) are more informative and related to health status and sports performance. Also, in the discussion section, a brief list of assessment methods (BIA, anthropometry, DXA) should be reported. See: Assessment of Body Composition in Athletes: A Narrative Review of Available Methods with Special Reference to Quantitative and Qualitative Bioimpedance Analysis. Nutrients. 2021 May 12;13(5):1620. doi: 10.3390/nu13051620. PMID: 34065984; PMCID: PMC8150618.
We have modified keywords, the introduction (lines 81-86) and the discussion (lines 294-297) and added a new reference [11].
The second paragraph is not really necessary and not linked with the 1st and 3rd. If you do not want to remove it, you can switch it with the 3rd.
“Then, many athletes report to consume foods with little nutritional value and not to 59 eat and drink sufficiently for practice and competition [13].” A sentence cannot be reported as a paragraph; please revise.
“Given these assumptions, athletes need more nutritional education to improve their 67 choices [8].” A sentence cannot be reported as a paragraph; please revise.
In general, there are too many paragraphs making the reading boring and not very concise.
We have revised the introduction reducing the paragraphs.
Methods: Study design should be the first paragraph.
We moved the “study design” paragraph as suggested.